# Reducing the effect of immortal time bias affects the analysis of prevention of delirium by suvorexant in critically ill patients: A retrospective cohort study

Junji Shiotsuka[1], Shigehiko Uchino[1], Yusuke Sasabuchi[2],
Tomoyuki Masuyama[1‡]*, Alan Kawarai Lefor[3‡], Masamitsu Sanui[1]

1 Department Anesthesiology and Critical Care Medicine, Saitama Medical Center, Jichi Medical University, Omiya, Saitama, Japan, 2 Data Science Center, Jichi Medical University, Shimotsuke, Tochigi, Japan, 3 Department of Surgery, Jichi Medical University, Shimotsuke, Tochigi, Japan

☯ These authors contributed equally to this work.
‡ TM and AKL also contributed equally to this work.
* tomomasuyama@gmail.com

**Data Availability Statement:** All relevant data are within the paper and its Supporting Information files.

## Abstract

### Background

Studies assessing the effect of suvorexant on delirium prevention included patients treated before development of delirium, which can introduce immortal time bias. The objective of the present study was to evaluate the effect of suvorexant on delirium, comparing patients treated before the onset of delirium with patients treated within 72h of admission using the same dataset.

### Methods

Data from adult patients admitted to the ICU from August 2018 to July 2021 were retrospectively analyzed. In "any time before" analysis, the incidence of delirium was compared for patients who received suvorexant at any time during their ICU stay (suvorexant) (unless delirium developed before treatment) with patients who either did not receive suvorexant or received suvorexant after development of delirium (control). This design was used in previously published studies. In "within 72h" analysis, the incidence of delirium was compared for patients who received suvorexant within 72 hours of admission (suvorexant) and patients who did not receive suvorexant or received it more than 72 hours after admission (control). Patients who developed delirium during the initial 72 hours were excluded from "within 72h" analysis (N = 799).

### Results

"Within 72h" analysis included 1,255 patients, and "any time before" analysis included 2,054 patients (of 6599 admissions). The unadjusted hazard ratio of "any time before" analysis was 0.16 and the 95% confidence interval was 0.13–0.21 (p<0.01). The adjusted hazard ratio was 0.21, and the 95% confidence interval was 0.16–0.27 (p<0.01). "Within 72h"

**Funding:** The author(s) received no specific funding for this work.

**Competing interests:** The authors have declared that no competing interests exist.

analysis had an unadjusted hazard ratio of 0.54 and the 95% confidence interval was 0.36–0.82 (p<0.01). However, this association lost statistical significance after adjustment for potential confounders (adjusted hazard ratio 1.02, 95% confidence interval 0.65–1.59, p = 0.93).

## Conclusion

Reducing the effect of immortal time bias led to a significantly reduced effect of suvorexant for the prevention of delirium.

## Introduction

Delirium is a common complication in hospitalized patients. The incidence of delirium is more common in the ICU than in general wards, and critically ill patients with delirium are more likely to have higher mortality and longer ICU and hospital lengths of stay [1]. Delirium also requires additional nursing resources, and is a risk factor for accidents in the ICU such as falls [2]. Therefore, it is not surprising that enormous efforts have been made to study the prevention of delirium including pharmacological strategies [3].

Orexins are two neuropeptides produced in the lateral hypothalamus, and play important roles in maintaining arousal stability. Dual orexin receptor antagonists predominantly enhance rapid eye movement sleep, which results in increasing total sleep time. Suvorexant is a dual orexin receptor antagonist approved as a sleep modulator in Japan and Australia since 2014 and has been widely used throughout the world. Suvorexant is approved in Japan for insomnia with an approved dose of 20mg for adults or 15mg for adults more than 65 years of age, once daily. Delirium prevention using suvorexant is a field of interest, since the sleep-wake disturbance has been regarded as one of the important risk factors for delirium [4]. To date, two small randomized controlled trials and nine observational studies have been published, and two systematic reviews and meta-analyses summarized the data [5–17]. The meta-analysis by Tian et al. included 2,594 participants in total and the effects on delirium were an odds ratio (OR) of 0.25 with a 95% confidence interval (CI) of 0.07 to 0.95 for two randomized controlled trials and an OR of 0.36 with the 95% CI of 0.23 to 0.56 for nine observational studies [17].

Although systematic reviews and meta-analyses showed beneficial effects of suvorexant on delirium prevention, the design of included studies was not optimal. Most of the previous observational studies did not clearly specify the beginning of the exposure, and thus a certain number of patients may not have received suvorexant at the beginning of the observation, although they were classified as "exposed" to suvorexant. If the beginning of the exposure is not specified explicitly, patients who had a primary outcome in the early stage of observation would tend to be classified into a control group, because preventive strategies are not used for those who already have the outcome. This phenomenon is referred to as "immortal time bias", which may result in finding an enhanced effect of a preventive strategy [18]. This makes it difficult to interpret evidence in previous studies regarding suvorexant which is summarized in meta-analyses that are prone to be biased.

The objective of the present study is to compare the preventive effect of suvorexant on delirium for patients in the ICU analyzed in two ways, one follows the method used in previously published observational studies ("any time before" analysis) and the other reduces the effect of

immortal time bias as much as possible ("within 72h" analysis) by explicitly stating the beginning of the exposure to suvorexant to be within 72h of admission to the ICU.

## Materials and methods

### Data source

In this retrospective cohort study, all required data was obtained from electronic medical records of patients admitted to the ICU (ACSYS, Phillips, Tokyo, Japan). The required data was abstracted electronically and did not involve manual review of medical records. The following data were collected: age, gender, height, weight, reason for ICU admission, acute physiology and chronic health evaluation (APACHE) II score at ICU admission, time-stamped drug prescriptions in the ICU, date of hospital admission and discharge, ICU admission and discharge time, start and end time of mechanical ventilation, diagnosis of delirium using the Confusion Assessment Method for the Intensive Care Unit (CAM-ICU), and discharge status. All patients in the ICU undergo standardized confusion assessment (CAM-ICU) for detecting delirium every 6 to 8 hours during their entire ICU stay by trained ICU nurses. The data collected also includes emergency calls or rapid response system (RRS) activation prior to ICU admission, location prior to ICU admission, and whether the patient underwent surgery. This study was approved prior to its commencement by Jichi Medical University, Saitama Medical Center, Institutional Review Board. Informed consents were waived by the regulation of the IRB, since the data were treated anonymously, and the opt-out information was uploaded on the web site of the institution (IRB approval No. S22-037).

### Patients

Patients admitted consecutively to the medical-surgical ICU at Jichi Medical University, Saitama Medical Center, Japan from 1 August 2017 through 31 July 2021 were included. Patients were considered eligible if they stayed in the ICU for at least 72 hours. The five exclusion criteria were: (i) patients discharged from the ICU within 72 hours (N = 3866), (ii) patients with disturbed level of consciousness (Glasgow Coma Scale <9 within 24 hours of admission to the ICU) (N = 433), (iii) patients who underwent neurological surgery (N = 69), (iv) those with any kind of brain injury such as cerebral hemorrhage, ischemic stroke, subarachnoid hemorrhage, traumatic brain injury, post-cardiac arrest syndrome, status epilepticus, or neurological infection (N = 132), and (v) patients who developed delirium on admission to the ICU (N = 25).

### Exposure and outcomes

Patients who received suvorexant within 72 hours of ICU admission or before the development of delirium were defined as the suvorexant group. Patients who did not receive suvorexant within 72 hours of ICU admission, or who received it after the development of delirium, were defined as the control group. The median time until developing delirium from the time of ICU admission in existing studies was longer than 3 days and therefore, 72 hours was selected as an appropriate time point for the interval of exposure to a preventive suvorexant use. Suvorexant was prescribed for patients in the ICU who complain of insomnia. The decision to administer suvorexant was made by intensivists in the ICU, or psychiatrists for some patients when consultations were obtained for patients' insomnia. All patients admitted to the ICU received non-pharmacological preventive strategies that were recommended in the guideline from the Society of Critical Care Medicine [19]. The primary outcome was the occurrence of delirium regardless of the type of delirium, such as hyperactive, hypoactive or mixed.

Although the primary outcome was set as a dichotomous occurrence of delirium, not a length of time of delirium nor a number of episodes of delirium during the ICU stay, there were multiple episodes in some patients. The occurrence of delirium was considered important for a study of preventive strategies, and previous studies also set the primary outcome as the dichotomous occurrence of delirium. Secondary outcomes were the onset of delirium, length of ICU stay, length of hospital stay, and death in the ICU or hospital.

## Other variables

Other variables evaluated included demographics (age and gender), comorbidity before ICU admission (maintenance hemodialysis, heart failure, and respiratory failure), emergency call before ICU admission, emergency admission to the ICU, location before ICU (general ward, operating room, or emergency department), type of diagnosis (non-surgical or surgical), APACHE II Score, treatment received within 24 hours of ICU admission (high flow nasal cannula (HFNC), continuous renal replacement therapy (CRRT), mechanical ventilation (MV), acute kidney injury (AKI)), and medications received within 24 hours of ICU admission (ramelteon, fentanyl, acetaminophen, dexmedetomidine, haloperidol, midazolam, propofol, quetiapine, steroids, or famotidine). An emergency call was categorized as no emergency call, activation of the rapid-response system, and emergency call. An admission category was classified as a non-surgical admission, an elective operation, or an emergency operation. Emergency admission to the ICU was defined as all ICU admissions except for scheduled operations.

## Statistical analysis

Numerical values were presented as means with standard deviations, or medians with interquartile ranges depending on their distribution. Categorical variables were presented as numbers and percentages. Differences in means were tested by Student's t-test and differences in medians were tested by the Wilcoxon rank-sum test. Differences in categorical variables were tested by the chi-squared test or Fisher's exact test depending on the number of values. Survival analyses were performed for the occurrence of delirium. The Cox regression model was made using age, APACHE II score, status post cardiovascular surgery, emergency admission, HFNC, CRRT, MV, AKI within 24 hours of admission, ramelteon, haloperidol, midazolam, propofol, tramadol, quetiapine, and corticosteroids as independent variables. These variables were selected due to their clinical importance and from previous studies. Proportional hazard assumption was assessed with Schoenfeld residuals. Stata version 16 (Stata Corp LLC, Texas, USA) was used for statistical analyses and p-values less than 0.05 were considered statistically significant.

## Sample size determination

The sample size was calculated using a power analysis at 166 patients for the exposure group and 542 patients for the control group with a power of 0.8 and an alpha of 0.05 with an expected hazard ratio of 0.47 with a censoring rate of 0.29 based on a previous study [13].

## Data analyses

Two separate analyses of the same cohort were performed to examine differences in results, one analysis performed in the manner used by previous studies ("any time before" analysis) and one that takes immortal time bias into account ("within 72h" analysis). Patient eligibility for both analyses was identical, but exclusion criteria and allocation were different. Both analyses had a suvorexant group and a control group.

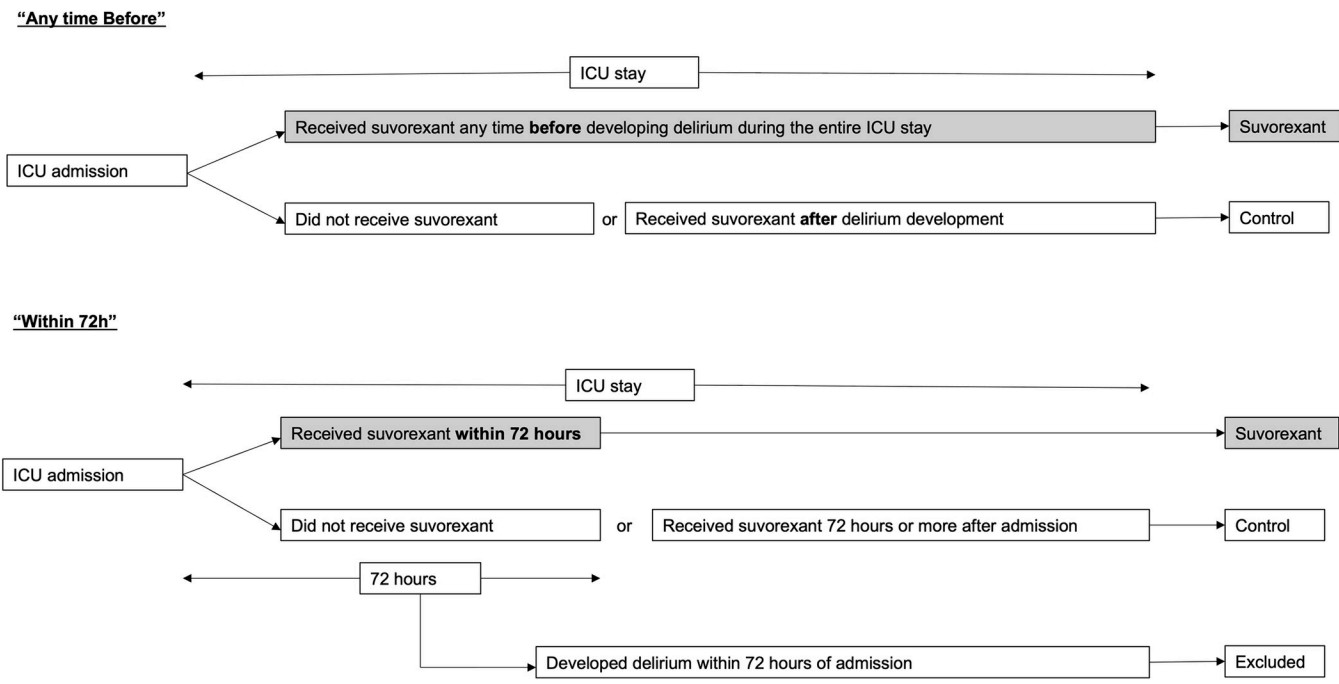

**Fig 1. Patient allocation flow. a.** Patient allocation flow: "Any time before" analysis. **b.** Patient allocation flow: "Within 72h" analysis.

In "any time before" analysis, the incidence of delirium was compared for patients who received suvorexant at any time during their ICU stay (suvorexant) (unless delirium developed before treatment) with patients who either did not receive suvorexant or received suvorexant after the development of delirium (control). In "within 72h" analysis, the incidence of delirium was compared for patients who received suvorexant within 72 hours of admission (suvorexant) with patients who did not receive suvorexant or received it more than 72 hours after admission (control). Patients who developed delirium during the initial 72 hours were excluded from "within 72h" analysis (N = 799).

All patients who received suvorexant during their ICU stay before the development of delirium were allocated to the suvorexant group in both analyses, and those who did not receive suvorexant or received it after the development of delirium were allocated to the control group (Fig 1A and 1B). There was a suvorexant group and a control group for each of the two analyses.

## Results

During the study period, there were 6599 patients consecutively admitted to the ICU. Of these, 4545 patients were excluded according to the five criteria above with 2054 patients remaining in the final analysis for this study (Fig 2A). Of these, 504 were treated with suvorexant before the development of delirium and 1550 did not receive suvorexant at all or received it after developing delirium.

### "Any time before" analysis

This analysis included all patients treated with suvorexant at any time before the development of delirium, in the manner of previous studies. There were 1550 patients in the control group who were not treated with suvorexant and 504 in the suvorexant group. Patient backgrounds

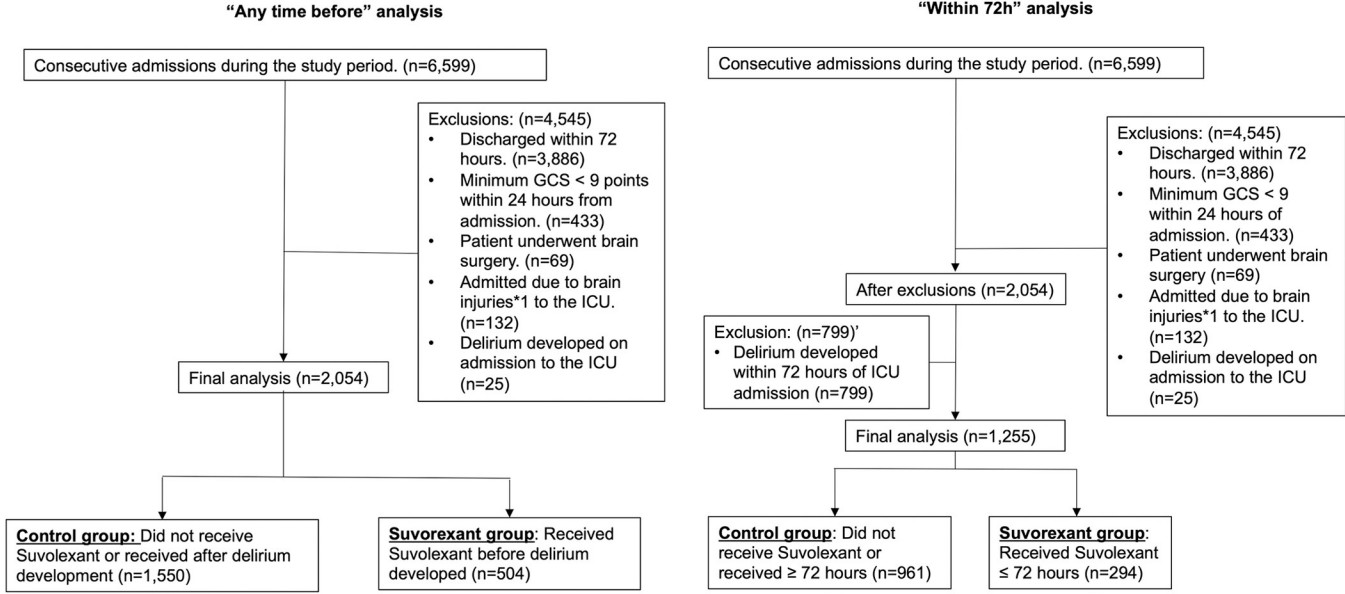

**Fig 2. Patient eligibility and exclusions. a.** Patient flow for the "any time before" analysis. **b.** Patient flow for the "within 72h" analysis.

in this analysis were somewhat heterogeneous (S1 Table). The suvorexant group was more likely to receive potentially confounding medications except for midazolam compared to the control group (S2 Table). Delirium developed in 911/1550 (58.8%) patients in the control group, and in 73/504 (14.5%) patients in the suvorexant group.

The number of patients who developed delirium was significantly lower in the suvorexant group. A proportional hazard assumption was confirmed by a plot of Schoenfield residuals (S1A Fig). The unadjusted HR and 95%CI were 0.16 and 0.13 to 0.21, respectively (Fig 3A: Kaplan-Meier survival curve, log-rank, p<0.01) The ventilator days were also shorter in the suvorexant group. However, the median ventilator days in the control group of this analysis was about two times longer than in the following "within 72h" analysis. Although the days to

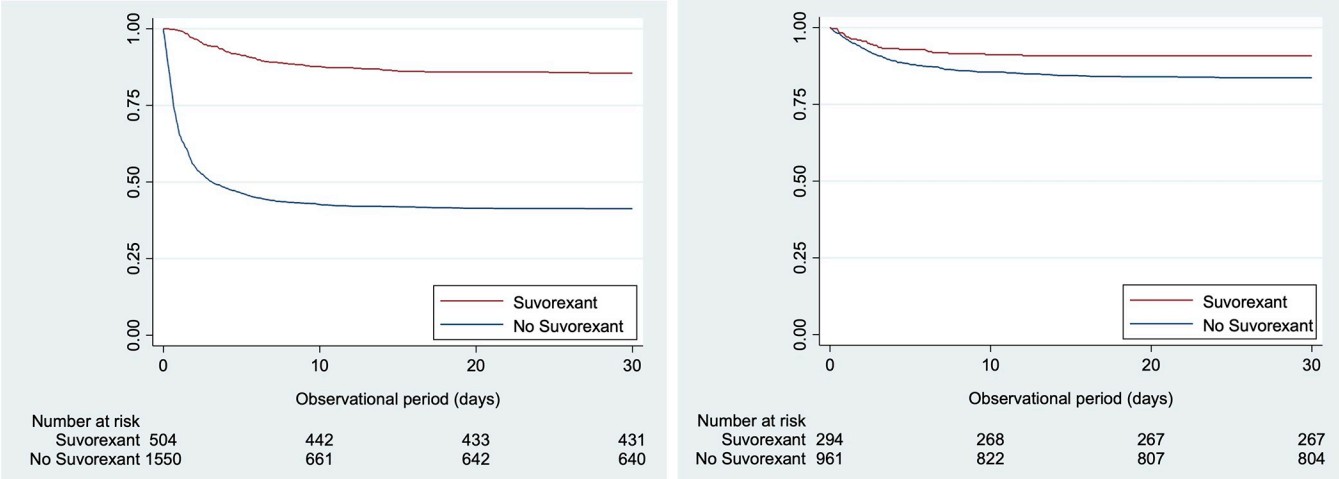

**Fig 3. Kaplan-Meier survival analysis. a.** Kaplan-Meier survival analysis of the "any time before" analysis. **b.** Kaplan-Meier survival analysis of the "within 72h" analysis.

Table 1. Effect of suvorexant on delirium adjusted by covariates in "any time before" analysis.

| Delirium | | p |
|---|---|---|
| *Suvorexant* | HR[d], 0.23; 95%CI[e], 0.18–0.30 | <0.01 |
| Age | HR, 1.02; 95%CI, 1.01–1.02 | <0.01 |
| APACHE2 Score | HR, 1.05; 95%CI, 1.03–1.06 | <0.01 |
| Status/post cardiovascular surgery | HR, 1.21; 95%CI, 0.98–1.50 | 0.08 |
| Emergency admission | HR, 1.03; 95%CI, 0.53–1.99 | 0.94 |
| HFNC[a] | HR, 1.33; 95%CI, 1.16–1.53 | <0.01 |
| CRRT[b] | HR, 1.17; 95%CI, 0.98–1.41 | 0.08 |
| AKI[c] within 24 hours from the admission | HR, 0.82; 95%CI, 0.65–1.04 | 0.11 |
| Ramelteon | HR, 0.66; 95%CI, 0.45–0.97 | 0.04 |
| Haloperidol | HR, 0.92; 95%CI, 0.67–1.25 | 0.60 |
| Midazolam | HR, 0.84; 95%CI, 0.67–1.07 | 0.16 |
| Tramadol | HR, 0.08; 95%CI, 0.05–0.15 | <0.01 |
| Quetiapine | HR, 1.15; 95%CI, 0.62–2.12 | 0.65 |
| Dexmedetomidine | HR, 0.81; 95%CI, 0.66–0.99 | 0.04 |

Effect of suvorexant on delirium adjusted by covariates in "any time before" analysis.

[a]HFNC: High Flow Nasal Cannula.

[b]CRRT: Continuous Renal Replacement Therapy.

[c]AKI: Acute Kidney Injury.

[d]HR: Hazard Ratio.

[e]CI: Confidence Interval.

the onset of the delirium from the admission to the ICU was not significantly different between two groups in the "within 72h" analysis, it was only 0.8 days in the control group and 3.9 days in the suvorexant group in this "any time before" analysis, which was significantly different. Both mortality at the ICU and the hospital were smaller in this "any time before" analysis (RR, 0.30; 95%CI, 0.18 to 0.52 for the ICU mortality, and RR, 0.35; 95%CI, 0.24 to 0.51 for the hospital mortality). The HR of the delirium was adjusted by age, APACHE II score, status post cardiovascular surgery, emergency admission, emergency call, admission categories, the place before admission to the ICU, use of high flow nasal cannula, use of continuous renal replacement therapy, acute kidney injury within 24 hours from the admission, and co-administered medicines, such as ramelteon, haloperidol, midazolam, tramadol, quetiapine, and dexmedetomidine. After adjustments for possible confounding factors, administration of suvorexant remained significantly associated with a decreased incidence of delirium (HR, 0.23; 95%CI, 0.18–0.30, p<0.01) (Table 1). This analysis was conducted in the manner of previous studies, with similar results, showing a significant reduction in the incidence of delirium in patients treated with suvorexant any time before the development of delirium.

## "Within 72h" analysis

Of the 2054 patients in the "any time before" analysis above, as an additional exclusion criterion, 799 patients who developed delirium within the first 72h of admission to the ICU were excluded from this analysis, to reduce the effect of immortal time bias (Fig 2B) by explicitly stating the beginning of the exposure to suvorexant and exclude patients who developed delirium during that time. The remaining 1,255 patients were divided into a control group of 961 patients who did not receive suvorexant or received it >72h after admission, and a suvorexant group, 294 patients who received suvorexant within the first 72h of ICU admission (Fig 2B). It

**Table 2. Patient characteristics.**

|  | Control (n = 961) | Suvorexant (n = 294) | p |
|---|---|---|---|
| Gender, Male | 645 (67.1) | 210 (71.4) | 0.17 |
| Age | 69 (59–76) | 71 (60–77) | 0.05 |
| APACHE2 | 17 (13–21) | 16 (13–20) | 0.03 |
| Maintenance HD[a] | 76 (7.9) | 31 (10.5) | 0.16 |
| Heart failure | 10 (1.0) | 2 (0.7) | 0.74 |
| Respiratory failure | 10 (1.0) | 1 (0.3) | 0.47 |
| Post cardiovascular surgery | 443 (46.1) | 193 (65.7) | <0.01 |
| Readmission | 56 (5.8) | 24 (8.2) | 0.15 |
| Emergency call; RRS[b]/ code blue | 91 (9.5) / 1 (0.1) | 21 (7.1) / 0 | 0.42 |
| Admission category; non-surgeries/ elective surgeries/ emergent surgeries | 403 (41.9)/ 387 (40.3) / 171 (17.8) | 84 (28.6)/ 174 (59.2) / 36 (12.2) | <0.01 |
| Emergency Admission | 574 (59.7) | 121 (41.2) | <0.01 |
| Place before the ICU; general ward/ OR[c]/ ED[d]/ others | 149 (15.5)/ 498 (51.8) / 227 (23.6) / 85 (8.8)/ 2 (0.2) | 33 (11.2)/ 196 (66.7) / 47 (16.0) / 18 (6.1)/ 0 | <0.01 |
| Days from admission to the first dose of suvorexant | 4.4 (3.4–7.2) | 1.5 (1.2–2.5) | |
|  | Control (n = 961) | Suvorexant (n = 294) | p |
| **Non-surgical admission** | | | |
| Cardiovascular disease | 147 (36.6) | 35 (42.2) | 0.34 |
| Respiratory disease | 127 (31.6) | 25 (30.1) | 0.79 |
| Gastrointestinal disease | 46 (11.4) | 8 (9.7) | 0.63 |
| Sepsis | 22 (5.4) | 6 (7.2) | 0.53 |
| Others | 60 (15.0) | 9 (10.8) | 0.33 |
| **Subtotal** | 402 | 83 | |
| **Surgical admission** | | | |
| Cardiovascular surgery | 443 (79.2) | 193 (91.5) | <0.01 |
| Pulmonary surgery | 28 (5.0) | 5 (2.4) | 0.11 |
| Gastrointestinal surgery | 48 (8.6) | 7 (3.3) | 0.01 |
| Others | 40 (7.2) | 6 (2.8) | 0.02 |
| **Subtotal** | 559 | 211 | |

Patient characteristics in "within 72h" analysis.

[a]HD: Hemodialysis.

[b]RRS: Rapid Response System.

OR: Operating Room.

[d]ED: Emergency Department.

is to be emphasized that the patients in the "within 72h" analysis were the same patients in the "any time before" analysis above, with the exclusion of 799 patients who developed delirium within 72h of admission to the ICU.

Patients in the suvorexant group were less severely ill than the control group assessed by the APACHE II score. More patients in the suvorexant group were admitted to the ICU after undergoing elective operations. Patients after cardiovascular surgery were more likely to be treated with suvorexant. Most admissions to the ICU were patients who had undergone cardiovascular surgery in both groups. (Table 2) With regard to events during their ICU stays, the suvorexant group more often required high flow nasal oxygen, whereas the control group more often required continuous renal replacement therapy and tended to suffer from acute kidney injury within 24 hours from their admission to the ICU (Table 3). Medications that

**Table 3. Other variables, co-administered medicines at the ICU, and outcomes.**

**Other variables**

| | Control (n = 961) | Suvorexant (n = 294) | p |
|---|---|---|---|
| High flow nasal oxygen | 264 (27.5) | 100 (34.1) | 0.03 |
| Non-invasive ventilation | 72 (7.5) | 29 (9.9) | 0.19 |
| Mechanical ventilation | 682 (71.0) | 223 (75.9) | 0.10 |
| Continuous renal replacement therapy | 175 (18.2) | 36 (12.2) | 0.02 |
| AKI[a] within 24 hours from the ICU admission | 60 (6.2) | 10 (3.4) | 0.06 |
| Urine output within 24 hours from the ICU admission | 1245 (740–1859) | 1266 (800–1940) | 0.64 |

**Co-administered medicines**

| | Control (n = 961) | Suvorexant (n = 294) | p |
|---|---|---|---|
| Fentanyl, continuous infusion | 633 (65.9) | 188 (64.0) | 0.54 |
| Tramadol with/without acetaminophen | 198 (20.6) | 85 (28.9) | <0.01 |
| Dexmedetomidine | 133 (13.8) | 47 (16.0) | 0.36 |
| Haloperidol | 51 (5.3) | 36 (12.2) | <0.01 |
| Midazolam | 89 (9.3) | 4 (1.4) | <0.01 |
| Propofol | 178 (18.5) | 41 (14.0) | 0.07 |
| Quetiapine | 15 (1.6) | 14 (4.8) | <0.01 |
| Ramelteon | 152 (15.8) | 162 (55.1) | <0.01 |
| Steroids | 118 (12.3) | 23 (7.8) | 0.03 |
| Famotidine | 62 (6.5) | 17 (5.8) | 0.68 |

**Outcomes**

| | Control (n = 961) | Suvorexant (n = 294) | p |
|---|---|---|---|
| Delirium | 158 (16.4) | 27 (9.2) | |
| | HR[b], 0.54; 95%CI[c], 0.36–0.82 | | <0.01 |
| Ventilator days (hours) | 30.3 (12.9–122.4) | 17.4 (8.7–67.4) | <0.01 |
| ICU length of stay (days) | 5.2 (3.9–8.3) | 5.2 (4.0–6.9) | 0.18 |
| Hosp length of stay (days) | 24 (17–40) | 25 (18–38) | 0.47 |
| Days to the onset of delirium from the admission to the ICU | 5.6 (4.2–8.4) | 5.3 (3.9–7.3) | 0.29 |
| Mortality at ICU discharge | 64 (6.7) | 4 (1.4) | <0.01 |
| | RR[d], 0.20; 95%CI, 0.08–0.57 | | <0.01 |
| Mortality at hospital discharge | 106 (11.1) | 12 (4.1) | <0.01 |
| | RR, 0.37; 95%CI, 0.21–0.66 | | <0.01 |

Other variables, co-administered medicines at the ICU, and outcomes in "within 72h" analysis.

[a]AKI: Acute Kidney Injury.

[b]HR: Hazard Ratio.

[c]CI: Confidence Interval.

[d]RR: Risk Ratio.

could have affected the incidence of delirium were summarized in Table 3. Tramadol with or without acetaminophen, haloperidol, ramelteon, and quetiapine were more often administered in the suvorexant group, whereas midazolam and any kinds of steroids were more likely to be administered in the control group. Ramelteon was given to about half of the patients administered suvorexant and was prescribed alone to very few patients.

Delirium occurred in 158/961 patients of the control group and in 27/294 patients in the suvorexant group (16.4% vs. 9.2%). As the Kaplan-Meier curve shows, administration of suvorexant was significantly associated with the delayed development of delirium (Fig 3B; log-rank test, p<0.01), and the unadjusted hazard ratio (HR) of the effect of suvorexant on

**Table 4. Effect of suvorexant on delirium adjusted by covariates.**

| Delirium | | p |
|---|---|---|
| *Suvorexant* | HR[d], 1.31; 95%CI[e], 0.79–2.19 | 0.30 |
| Age | HR, 1.02; 95%CI, 1.01–1.04 | <0.01 |
| APACHE2 Score | HR, 1.03; 95%CI, 1.00–1.06 | 0.03 |
| Status/post cardiovascular surgery | HR, 1.13; 95%CI, 0.75–1.70 | 0.56 |
| Emergency admission | HR, 4.69; 95%CI, 2.84–7.75 | <0.01 |
| HFNC[a] | HR, 1.65; 95%CI, 1.20–2.27 | <0.01 |
| CRRT[b] | HR, 1.76; 95%CI, 1.22–2.55 | <0.03 |
| Mechanical ventilation | HR, 2.58; 95%CI, 1.55–4.28 | <0.01 |
| AKI[c] within 24 hours from the admission | HR, 0.66; 95%CI, 0.37–1.18 | 0.16 |
| Ramelteon | HR, 0.58; 95%CI, 0.29–1.18 | 0.13 |
| Haloperidol | HR, 1.96; 95%CI, 1.24–3.10 | <0.01 |
| Midazolam | HR, 2.44; 95%CI, 1.65–3.63 | <0.01 |
| Propofol | HR, 2.10; 95%CI, 1.44–3.06 | <0.01 |
| Tramadol | HR, 0.14; 95%CI, 0.05–0.38 | <0.01 |
| Quetiapine | HR, 1.41; 95%CI, 0.64–3.09 | 0.39 |
| Steroids | HR, 0.66; 95%CI, 0.44–1.01 | 0.06 |

Effect of suvorexant on delirium adjusted by covariates in the "Within 72h" analysis.

[a]HFNC: High Flow Nasal Cannula.

[b]CRRT: Continuous Renal Replacement Therapy.

[c]AKI: Acute Kidney Injury.

[d]HR: Hazard Ratio.

[e]CI: Confidence Interval.

delirium was 0.54 and 95%CI was 0.36–0.82 (p<0.01). Ventilator-days were shorter in the exposure group. Mortalities at ICU discharge and at hospital discharge were lower in the exposure group (Table 3).

As a proportional hazard assumption was confirmed by a plot of Schoenfield residuals (S1B Fig), a Cox regression model was developed including age, APACHE II score, use of mechanical ventilation, use of high flow nasal oxygen, use of continuous renal replacement therapy, the occurrence of acute kidney injury within 24 hours from admission, status post cardiovascular surgeries, the admission category, the emergency admission, the location before admission to the ICU, and co-administered medicines, such as ramelteon, fentanyl, tramadol, midazolam, propofol, quetiapine, steroid, and famotidine. The significant association of a decreased incidence of delirium with administration of suvorexant which was found before adjustment for possible confounding factors was eliminated after adjustment for these possible confounding factors (HR, 1.31 with 95%CI of 0.79–2.19, p = 0.30) (Table 4). This analysis reduces the effect of immortal time bias using the data from the same patient cohort as "any time before" analysis, by explicitly stating the beginning of the exposure to suvorexant.

## Discussion

In the present study, the association of suvorexant with the occurrence of delirium was evaluated in the ICU using two different methods of analysis. In "any time before" analysis, patients were analyzed as exposed if they were administered suvorexant at any time during their stay in the ICU unless suvorexant was administered after the development of delirium, consistent with the analysis in previous studies of suvorexant. Patients who were given suvorexant after

the development of delirium were allocated to the control group in the "any time before" analysis. In "within 72h" analysis, the period of exposure is explicitly stated at the beginning of the observation to be 72h. If patients developed delirium during this predefined period, they were excluded from further analysis. Although the unadjusted HRs were toward a significant decrease in delirium in patients treated with suvorexant, the adjusted HR did not have a significant association between administration of suvorexant and a decreased incidence of delirium in the "within 72h" analysis, whereas the adjusted HR was still significant association between suvorexant use and a decreased incidence of delirium in the "any time before" analysis.

The effect of suvorexant on the development of delirium was reported in two randomized controlled trials and nine observational studies [5–13, 15, 20]. A stronger association between suvorexant and delirium was observed in nine observational studies with 2,542 patients compared to the two randomized controlled trials with 140 patients. Four studies divided the patients into groups chronologically to confirm the effect of suvorexant [6, 11, 14, 15]. One study observed a group of patients for a certain period of time and assumed an association between suvorexant use and the incidence of delirium, which was a cross-sectional design. However, eight retrospective observational studies were cross-sectional studies, because these studies did not make a clear distinction between the baseline period and the outcome observation period [18]. The study designs used in the eight retrospective observational studies were of concern due to the effect of immortal time bias, toward enhancing the apparent effect of suvorexant on the incidence of delirium. Patients who develop the outcome of interest in the early stage of an observational period are less likely to be allocated to the exposure group, resulting in improved outcomes in the exposure group. The effects of suvorexant in the "any time before" analysis are similar to that reported in the previous studies. The effect size of the HR in the "any time before" analysis was greater than that in the "within 72h" analysis.

Patients admitted to the ICU share more risk factors for delirium than patients in general wards, such as monitor noises, lights in the middle of the night, disease severity, medical equipment, and so on [21]. This could result in diluting the effect of suvorexant as well as enhancing its effect if the study is not optimally designed. Data regarding time to develop delirium from the time of admission to the ICU in studies testing the preventive effect of suvorexant on delirium are not clearly presented. Masuyama et al. reported the median time and the interquartile range as 3.81 days and 1.81 to 5.79 days in overall, 5.06 days and 3.08 to 8.81 days in the exposure group, and 3.59 days and 1.59 to 4.90 days in the control group in the ICU [8]. Rahimibashar et al. reported the mean time to the recognition of delirium as 7.55 ± 1.88 days in their study conducted in two teaching-hospital ICUs [22]. Based on Kaplan-Meier curves reported in previous studies, the median time to the onset of delirium from the time of ICU admission is about 3 to 5 days regardless of the study location [5–9, 13]. In the "within 72h" analysis of the present study, the median time to development of delirium was 5 to 6 days in both the control and suvorexant groups, whereas in the "any time before" analysis, the median time to development of delirium was 0.8 days in the control group and 3.9 days in the suvorexant group, which seemed to be shorter than previously reported partly due to the illness severity of patients admitted to this ICU. This could have resulted from a type of the delirium not investigated in the present study, as well as in previously reported studies. This result definitely reflects the influence of immortal time bias since patients who developed delirium earlier would no longer receive prevention, and this difference would have been observed if previously reported studies had been reanalyzed as done in the present study. However, this may have enhanced the effect of suvorexant in the "any time before" analysis.

There are acknowledged limitations to this study. First, as a retrospective study, it is not possible to control the timing of and the reasons for administering suvorexant. Since some clinicians involved in the current study had reported the effect of suvorexant previously, this

could affect clinical decision-making, although the decisions were solely at each physician's discretion [8]. Nurses in the ICU may have requested orders for suvorexant for patients they felt were at high risk for developing delirium, even though the patient did not complain of insomnia. Second, there were significant background diversities between the groups in both analyses, such as types of surgery and co-administered medications that were not able to be fully adjusted statistically. If these diversities contribute to enhance the preventive effect of suvorexant on delirium, it is difficult to apply the results to clinical practice. However, the present study demonstrates the disappearance of the effect of suvorexant in analyses with reduced biases, and it is suggested that these diversities do not cause severe problems, although the results should be interpreted with caution. Third, patients admitted to the ICU with non-surgical problems were generally more severely ill than those with surgical conditions, and they were more likely intubated with some degree of sedation and analgesics. Therefore, they did not receive suvorexant in the first several days and were allocated to the control group. These facts may have led to differences between the two analyses. Fourth, the diagnoses of delirium were made solely by CAM-ICU obtained by trained nurses. Since the analyzed data were automatically retrieved from the electronic medical record, nurse charting on the patients record are the only reliable and available data for the diagnosis. CAM-ICU was originally developed by Ely, et al. in 2001 and reported as the sensitive of 93% to 100% and the specificity of 98% to 100% compared with the assessments by geriatricians [23]. The pooled sensitivity and the pooled specificity were reported later on as 80% and 96%, respectively [24]. This relatively low sensitivity yet high specificity could affect the results. However, many previous delirium studies published in the 2000s used CAM-ICU for diagnoses of delirium, and as long as practice in this ICU is based on CAM-ICU, the relatively low sensitivity is not considered to be a significant problem. A large-scale randomized controlled trial is warranted to confirm the results of the present study. This study was from a single center in Japan. Since treatment administered in ICUs differ among facilities, caution is necessary regarding generalizability.

## Conclusion

In conclusion, the present study shows that reducing the effect of immortal time bias led to a significantly reduced effect of suvorexant for the prevention of delirium. These results suggest that failure to do this in previous studies may have led to analyses showing an inflated effect of suvorexant for the prevention of delirium. Although several meta-analyses have been published, caution is needed when interpreting these studies.

## Supporting information

**S1 Fig. Schoenfield residuals. a.** Schoenfield residuals of the "any time before" analysis. **b.** Schoenfield residuals of the "within 72h" analysis.
(ZIP)

**S1 Table. Patients' backgrounds.** Patients' backgrounds in the "any time before" analysis.
(DOCX)

**S2 Table. Other variables, co-administered medicines at the ICU, and the outcomes.** Other variables, co-administered medicines at the ICU, and the outcomes in "any time before" analysis.
(DOCX)

**S1 Data. Data set to reach the conclusion.** Data set that was used to reach the conclusion.
(XLSX)

**S1 File. Explanations of variables in the data set.** Explanations of variables that were used to analyze in the present study.
(DOCX)

## Author Contributions

**Conceptualization:** Junji Shiotsuka.

**Data curation:** Junji Shiotsuka, Shigehiko Uchino.

**Formal analysis:** Junji Shiotsuka, Yusuke Sasabuchi, Alan Kawarai Lefor.

**Methodology:** Junji Shiotsuka, Shigehiko Uchino, Yusuke Sasabuchi, Tomoyuki Masuyama, Alan Kawarai Lefor, Masamitsu Sanui.

**Writing – original draft:** Junji Shiotsuka.

**Writing – review & editing:** Junji Shiotsuka, Shigehiko Uchino, Yusuke Sasabuchi, Tomoyuki Masuyama, Alan Kawarai Lefor, Masamitsu Sanui.

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
