## [Decision Letter · Decision Letter 0]

5 Oct 2022

PONE-D-22-21193Reducing the effect of immortal time bias affects the analysis of prevention of delirium by suvorexant: A retrospective cohort studyPLOS ONE

Dear Dr. Masuyama,

Thank you for submitting your manuscript to PLOS ONE. After careful consideration, we feel that it has merit but does not fully meet PLOS ONE’s publication criteria as it currently stands. Therefore, we invite you to submit a revised version of the manuscript that addresses the points raised during the review process.

ACADEMIC EDITOR: *Your manuscript has been rigorously reviewed. Please consider editing and justification point-by-point according to the reviewer's recommendations.*

We look forward to receiving your revised manuscript.

Kind regards,

Konlawij Trongtrakul, MD, PhD

Academic Editor

PLOS ONE

Journal Requirements:

Additional Editor Comments:

- Please provide why using 72 hours of ICU admission or before the development of delirium as the suvorexant group in immortal time bias analysis.

- Additionally, several issues need to be corrected or clarified as per the reviewer's recommendations as follows:

Reviewers' comments:

Reviewer's Responses to Questions

**Comments to the Author**

1. Is the manuscript technically sound, and do the data support the conclusions?

Reviewer #1: Yes

Reviewer #2: Partly

2. Has the statistical analysis been performed appropriately and rigorously? 

Reviewer #1: Yes

Reviewer #2: Yes

3. Have the authors made all data underlying the findings in their manuscript fully available?

Reviewer #1: Yes

Reviewer #2: Yes

4. Is the manuscript presented in an intelligible fashion and written in standard English?

Reviewer #1: Yes

Reviewer #2: Yes

5. Review Comments to the Author

Reviewer #1: This is quite interesting study regarding the effect of suvorexant for delirium prevention in ICU. The study is well designed and good methodology. However. I still have some questions regarding this study.

1. This study is quite difficult to understand. It is probably be their terminology. " the immortal time bias" "anytime before" " within 72 hr"

2. The author should described which program they use for statistical analysis? The authors should add " p value < 0.05 is considered statistically significant.

2. Regarding the result of study. Table 1. the authors described there was statistically significant different between types of surgical procedures in patients with and without suvorexant. However, they should describe which type of surgery was different.

Table 2. Co administered medication; Patients in suvorexant received significantly higher tramadol, haloperidol, ramelteon. They did not explain why?

The outcome: Delirium. The authors demonstrated both OR and HR. I think the authors had better report only RR or HR. If they demonstrated both they should explain the different between RR and HR

Reviewer #2: Reducing the effect of immortal time bias affects the analysis of prevention of delirium by suvorexant: A retrospective cohort study

Comments

This is an interesting study in term of the study design to lessen the immortal time bias for the effect of suvorexant on the occurrence of delirium. Even the favorable outcomes of suvorexant on the incidence of delirium was addressed in previous studies but some limitations regarding the time of suvorexant administration was not clearly identified. I am impressed for the concept of this study; however, the retrospective cohort design has resulted in several limitation for interpretation of the outcomes. Given that, the imbalance in baseline characteristics between control and exposed groups would be the main concern.

Several comments need to be delineated.

• Title :

- I recommend to add “ in critically ill patients” at the end of the title

• Introduction:

- Please add reference for the sentence in line 67,69,76

- Line 77 “summarized” (past tense)

- Line 97, please provide the rationale for using the time frame “within 72 h” ,

any other previous studies mentioned about the onset of delirium in Japanese ICUs?

• Methods:

-If delirium had occurred several time during ICU stay, the investigator would collect only the first time?

- Although it was mentioned in the limitation, the author should give more information regarding the general indications and the dosage of suvorexant because suvorexant is not available in many countries. Any other indication than insomnia? Who have ordered? Intensivist? Psychiatrist? Geriatrician? This issue might interfere with the outcomes.

- Please provide the data regarding the accuracy of CAM-ICU (sens & spec) in practice because there were studies reported lower accuracy for CAM-ICU in practice.

- Did you have any data about the types of delirium (hypo,hyper,mixed) and other non-pharmacologic intervention (ABCDEF bundles) for prevention of delirium? Because non-pharmacologic intervention showed more effectively in prevention of delirium in ICU than pharmacologic prevention. Did you control for this intervention between groups?

- Please provide information about sample size calculation

• Results:

-There was significantly different in baseline characteristics between 2 groups in both analyses. This issue should be mentioned in limitation section. Patients in suvorexant showed higher number of surgical patients especially post cardiac surgery that seem to be less severity than control group.

-The onset of delirium was only 0.8 day in control group? Was it hypo or hyperactive delirium? If it was hypoactive delirium, it might result from the anesthesia -related effect?

- Could you explain whether ramelteon wasn’t adjusted in both analyses? Ramelteon is a melatonin receptor agonist that might improve sleep quality and decrease delirium in ICUs (even still inconclusive)

- Please provide the title of all figures

• Discussion:

- Please add detail in limitation section as mentioned above

- Limitations, consider word “first” because second, third,… were not mentioned?

Please re-check English grammar

6. PLOS authors have the option to publish the peer review history of their article (what does this mean?). If published, this will include your full peer review and any attached files.

Reviewer #1: No

Reviewer #2: **Yes: **ONUMA CHAIWAT

---

## [Author Response · Author response to Decision Letter 0]

16 Oct 2022

Editor Comments:

- Please provide why using 72 hours of ICU admission or before the development of delirium as the suvorexant group in immortal time bias analysis.

We added an explanation to explain the rationale of within “72 hours of ICU admission”. This important consideration is based on existing literature and published results. There are no set “standards” for this determination. Please see lines 125 to 127.

Comments to the Author

Reviewer #1: This is quite interesting study regarding the effect of suvorexant for delirium prevention in ICU. The study is well designed and good methodology. However. I still have some questions regarding this study.

1. This study is quite difficult to understand. It is probably be their terminology. " the immortal time bias" "anytime before" " within 72 hr"

We apologize for any difficulties in understanding. We have revised the beginning of the abstract where these terms were not used consistently with the rest of the paper. The term “immortal time bias”, we believe, is a standard term. We believe that we have used it correctly throughout the paper. In the Methods section, we explain the two types of analysis, the “any time before” analysis and the “within 72h” analysis, in the expanded “Data Analysis” section which clearly explains that 2 analyses were performed on the same patient cohort, and each analysis compared a suvorexant group with a control group. Please see lines 180 – 188.

2. The author should described which program they use for statistical analysis? The authors should add " p value < 0.05 is considered statistically significant.

We added these descriptions. Please see lines 167 – 168.

2. Regarding the result of study. Table 1. the authors described there was statistically significant different between types of surgical procedures in patients with and without suvorexant. However, they should describe which type of surgery was different.

We calculated the chi-squared analysis for each operative procedure and added the p-value to the table. Please see Table 1 and S1 Table.

Table 2. Co administered medication; Patients in suvorexant received significantly higher tramadol, haloperidol, ramelteon. They did not explain why?

Because we could not control each physician’s medical practice due to the retrospective design, we can not explain the reason for these differences. We acknowledge the differences as a limitation and describe this in the revised discussion section. Please see line 383 – 389.

The outcome: Delirium. The authors demonstrated both OR and HR. I think the authors had better report only RR or HR. If they demonstrated both they should explain the different between RR and HR

We eliminated the RR and kept HR in the table. Please see Table 2 and S2 table.

 

Reviewer #2: Reducing the effect of immortal time bias affects the analysis of prevention of delirium by suvorexant: A retrospective cohort study

Comments

This is an interesting study in term of the study design to lessen the immortal time bias for the effect of suvorexant on the occurrence of delirium. Even the favorable outcomes of suvorexant on the incidence of delirium was addressed in previous studies but some limitations regarding the time of suvorexant administration was not clearly identified. I am impressed for the concept of this study; however, the retrospective cohort design has resulted in several limitation for interpretation of the outcomes. Given that, the imbalance in baseline characteristics between control and exposed groups would be the main concern.

Several comments need to be delineated.

• Title :

- I recommend to add “ in critically ill patients” at the end of the title

We added “in critically ill patients” to the title. Please see line 2.

• Introduction:

- Please add reference for the sentence in line 67,69,76

- Line 77 “summarized” (past tense)

- Line 97, please provide the rationale for using the time frame “within 72 h” ,

any other previous studies mentioned about the onset of delirium in Japanese ICUs?

We added the references to the lines as Reviewer #2 kindly pointed out. Please see lines 58, 59 and 67 (because of an addition for other reviewer’s comment, the line number was changed). The rationale for 72hr is explained in lines 125-127, and we responded to a similar comment above from the editor.

• Methods:

-If delirium had occurred several time during ICU stay, the investigator would collect only the first time?

As this was research for prevention and previous studies also investigated the “binary” outcome of delirium, we did not take into consideration multiple events of delirium. Yes, data was collected only for the first time. We added an explanation in lines 133 – 138.

- Although it was mentioned in the limitation, the author should give more information regarding the general indications and the dosage of suvorexant because suvorexant is not available in many countries. Any other indication than insomnia? Who have ordered? Intensivist? Psychiatrist? Geriatrician? This issue might interfere with the outcomes.

As for the indication and doses, we added the information to the Background section. Please see line 64 – 65.

As for the person who decides the prescription, this was added to the methods section. Please see line 128 – 130.

- Please provide the data regarding the accuracy of CAM-ICU (sens & spec) in practice because there were studies reported lower accuracy for CAM-ICU in practice.

We added information regarding CAM-ICU to the revised discussion section. Please see lines 393 – 402.

- Did you have any data about the types of delirium (hypo,hyper,mixed) and other non-pharmacologic intervention (ABCDEF bundles) for prevention of delirium? Because non-pharmacologic intervention showed more effectively in prevention of delirium in ICU than pharmacologic prevention. Did you control for this intervention between groups?

As for the types of delirium, the definition of delirium in this study was based on CAM-ICU, and, therefore, we do not have data about the types of delirium. Please see line 132 – 133 and line 371 – 372. As for the non-pharmacological prevention, we added information. Please see lines 130 – 132.

- Please provide information about sample size calculation

Please see lines 170 – 173 in the new section “Sample Size Determination”.

• Results:

-There was significantly different in baseline characteristics between 2 groups in both analyses. This issue should be mentioned in limitation section. Patients in suvorexant showed higher number of surgical patients especially post cardiac surgery that seem to be less severity than control group.

We added this information to the limitation section. Please see line 389 – 393.

-The onset of delirium was only 0.8 day in control group? Was it hypo or hyperactive delirium? If it was hypoactive delirium, it might result from the anesthesia -related effect?

As we mentioned earlier, we cannot know if this difference caused by the types of delirium. However, we added the possible reason and explanation to the revised Discussion section. Please see line 371 – 375.

- Could you explain whether ramelteon wasn’t adjusted in both analyses? Ramelteon is a melatonin receptor agonist that might improve sleep quality and decrease delirium in ICUs (even still inconclusive)

We initially thought that most administration of ramelteon was made with suvorexant, there was no need to adjust for it. We reanalyzed with ramelteon and found a small difference. Please see tables 3a and 3b. 

However, we apologize to the editors and reviewers with regard to the mis-transcription of the data to table 3b. The hazard ratio of emergency admission was smaller than the real number. The HR became larger in this revision, which is not due to the adjustment for ramelteon.

- Please provide the title of all figures

We added titles for each figure to the figure legends.

• Discussion:

- Please add detail in limitation section as mentioned above

Please see above. We have added considerable detail to the limitation section.

- Limitations, consider word “first” because second, third,… were not mentioned?

We added several limitations and, thus, we changed the description. Please see lines 344 – 371.

Please re-check English grammar

We greatly appreciate your review and comment. This manuscript was thoroughly reviewed and revised by one of the co-authors, Professor Alan Kawarai Lefor who is a native English speaker and well-published surgical scientist. While there may be an occasional typographic error or stylistic difference, we believe that this manuscript conforms to standard scientific English language usage. We would certainly be willing to address any specific issues that you identify.

---

## [Decision Letter · Decision Letter 1]

6 Nov 2022

Reducing the effect of immortal time bias affects the analysis of prevention of delirium by suvorexant in critically ill patients: A retrospective cohort study

PONE-D-22-21193R1

Dear Dr. Masuyama,

We’re pleased to inform you that your manuscript has been judged scientifically suitable for publication and will be formally accepted for publication once it meets all outstanding technical requirements.

Kind regards,

Konlawij Trongtrakul, MD, PhD

Academic Editor

PLOS ONE

Additional Editor Comments (optional):

Reviewers' comments:

Reviewer's Responses to Questions

**Comments to the Author**

1. If the authors have adequately addressed your comments raised in a previous round of review and you feel that this manuscript is now acceptable for publication, you may indicate that here to bypass the “Comments to the Author” section, enter your conflict of interest statement in the “Confidential to Editor” section, and submit your "Accept" recommendation.

Reviewer #1: All comments have been addressed

2. Is the manuscript technically sound, and do the data support the conclusions?

Reviewer #1: Yes

3. Has the statistical analysis been performed appropriately and rigorously? 

Reviewer #1: Yes

4. Have the authors made all data underlying the findings in their manuscript fully available?

Reviewer #1: Yes

5. Is the manuscript presented in an intelligible fashion and written in standard English?

Reviewer #1: Yes

6. Review Comments to the Author

Reviewer #1: (No Response)

7. PLOS authors have the option to publish the peer review history of their article (what does this mean?). If published, this will include your full peer review and any attached files.

Reviewer #1: No

---

## [Editor Report · Acceptance letter]

18 Nov 2022

PONE-D-22-21193R1 

Reducing the effect of immortal time bias affects the analysis of prevention of delirium by suvorexant in critically ill patients: A retrospective cohort study 

Dear Dr. Masuyama:

I'm pleased to inform you that your manuscript has been deemed suitable for publication in PLOS ONE. Congratulations! Your manuscript is now with our production department. 

Kind regards, 

on behalf of

Associate Professor Konlawij Trongtrakul 

Academic Editor

PLOS ONE